# Dietary Intakes and Eating Behavior between Metabolically Healthy and Unhealthy Obesity Phenotypes in Asian Children and Adolescents

**DOI:** 10.3390/nu14224796

**Published:** 2022-11-12

**Authors:** Delicia Shu Qin Ooi, Jia Ying Toh, Lucas Yan Bin Ng, Zikang Peng, Supeng Yang, Nurul Syafiqah Binte Said Abdul Rashid, Andrew Anjian Sng, Yiong Huak Chan, Mary Foong-Fong Chong, Yung Seng Lee

**Affiliations:** 1Department of Paediatrics, Yong Loo Lin School of Medicine, National University of Singapore, Singapore 117549, Singapore; 2Khoo Teck Puat-National University Children’s Medical Institute, National University Health System, Singapore 119074, Singapore; 3Singapore Institute for Clinical Sciences, Agency for Science, Technology and Research, Singapore 117609, Singapore; 4Hwa Chong Institution, Singapore 269734, Singapore; 5Biostatistics Unit, Yong Loo Lin School Medicine, National University of Singapore, Singapore 117549, Singapore; 6Saw Swee Hock School of Public Health, National University of Singapore, Singapore 117549, Singapore

**Keywords:** dietary intakes, eating behavior, metabolically healthy obesity, children with obesity, adolescents with obesity

## Abstract

Diet plays a critical role in the development of obesity and obesity-related morbidities. Our study aimed to evaluate the dietary food groups, nutrient intakes and eating behaviors of metabolically healthy and unhealthy obesity phenotypes in an Asian cohort of children and adolescents. Participants (*n* = 52) were asked to record their diet using a 3-day food diary and intakes were analyzed using a nutrient software. Eating behavior was assessed using a validated questionnaire. Metabolically healthy obesity (MHO) or metabolically unhealthy obesity (MUO) were defined based on criteria of metabolic syndrome. Children/adolescents with MUO consumed fewer whole grains (median: 0.00 (interquartile range: 0.00–0.00 g) vs. 18.5 g (0.00–69.8 g)) and less polyunsaturated fat (6.26% kcal (5.17–7.45% kcal) vs. 6.92% kcal (5.85–9.02% kcal)), and had lower cognitive dietary restraint (15.0 (13.0–17.0) vs. 16.0 (14.0–19.0)) compared to children/adolescents with MHO. Deep fried food, fast food and processed convenience food were positively associated with both systolic (β: 2.84, 95%CI: 0.95–6.62) and diastolic blood pressure (β: 4.83, 95%CI: 0.61–9.04). Higher polyunsaturated fat intake (OR: 0.529, 95%CI: 0.284–0.986) and cognitive dietary restraint (OR: 0.681, 95%CI: 0.472–0.984) were associated with a lower risk of the MUO phenotype. A healthier diet composition and positive eating behavior may contribute to favorable metabolic outcomes in children and adolescents with obesity.

## 1. Introduction

Childhood obesity is one of the most pertinent public health challenges worldwide [1]. According to the World Health Organization, over 340 million children and adolescents aged 5–19 were overweight or obese in 2016, with childhood obesity levels reaching epidemic levels in developed countries [2]. Children and adolescents with obesity are at higher risks of reduced quality of life and lower life expectancy as excessive adiposity is the root cause of debilitating metabolic diseases including insulin resistance, hypertension and dyslipidemia [3].

Metabolic health in obesity can be heterogeneous, where not all children with obesity manifest adverse metabolic abnormalities. The subset of children with obesity but who do not present with metabolic abnormalities are classified as having “metabolically healthy obesity” (MHO) [4,5,6]. By definition, MHO presents a more favorable metabolic profile of higher levels of high-density lipoprotein (HDL) cholesterol and lower levels of blood pressure, fasting triglycerides and fasting glucose compared to their counterpart, “metabolically unhealthy obesity” (MUO) [7,8,9]. In addition, children with MHO have been shown to be younger, and have a lower waist to hip ratio, body mass index (BMI) percentile and body fat percentage [10,11,12,13].

Obesity is influenced by genetic, behavioral and obesogenic environmental factors such as an unhealthy diet and lack of physical activity [14]. In particular, diet is considered a major contributor of obesity and its comorbidities, and is often an important target for intervention strategies in the treatment of obesity and obesity-related morbidities [15]. Greater dietary intakes of glucose and trans fat were shown to be associated with increased risk of obesity and metabolic syndrome [16]. Higher total energy and total fat intake were associated with the MUO phenotype [17], while a higher intake of dietary fiber was found to be a predictor of the MHO phenotype in children [18]. The adherence to a Mediterranean diet, characterized by a high intake of vegetables, fruits, nuts, beans, whole grains and fish, was significantly higher in children with MHO compared to children with MUO [19]. In a study by Camhi et al., children with MHO were found to have a higher healthy eating index compared to children with MUO, and the difference was contributed to by a lower consumption of solid fats and added sugars among the children with MHO [20]. Studies have also demonstrated that children with MUO consume more sugar-sweetened beverages (SSB), salty snacks and fast food than children with MHO [21,22,23].

Apart from diet, eating behaviors are important in influencing energy balance (positive or negative), which is associated with obesity [24]. Rigid control, disinhibition and emotional susceptibility in eating behaviors were shown to be positively correlated to BMI z-scores among adolescents [25]. Moreover, subjects with metabolic syndrome were shown to display poorer eating behaviors including higher motivation to eat, higher emotional eating and a higher perception gap about feelings of fullness and hunger [26]. Adults with MHO were found to have a lower tendency to overeat when stressed compared to adults with MUO [27].

Since diet and eating behavior are modifiable risk factors, the comparison of dietary intakes and eating behavior between MHO and MUO phenotypes may highlight the nutritional components and food approach practices that can be altered to achieve better metabolic health among individuals with obesity. The reported studies on the MHO and MUO phenotypes in children are mainly based on American and European populations [19,20,21,22,23], who have different dietary intakes compared to Asian populations [28]. Hence, this study aimed to compare the intakes of food groups and nutrients, and eating behavior between metabolically healthy and unhealthy obesity phenotypes in an Asian cohort of children and adolescents.

## 2. Materials and Methods

### 2.1. Study Participants

Children and adolescents with obesity (*n* = 52) included in this study were from the OBesity in Singapore Children (OBiSC) study and they were of Chinese, Malay and Indian ethnicity. The participants were recruited from National University Hospital (NUH) and Health Promotion Board (HPB), Singapore. The recruitment criteria for these children and adolescents with obesity, and aged 7 to 19 years old were: (1) obese before age of 10 years, (2) BMI for age ≥97th percentile, (3) no syndromic causes of obesity. Medical examinations and history were obtained during study visits. The study was performed in accordance with the Declaration of Helsinki and ethics approval was obtained from Domain Specific Review Board of National Healthcare Group, Singapore (reference number: 2015/00314). Written informed consent was obtained from all study participants and their parents or legal guardian. The study is registered under clinicaltrials.gov (NCT02418377).

### 2.2. Anthropometric and Biochemical Measurements

Standard anthropometric parameters including weight, height, waist and hip circumference were measured. BMI was calculated as weight in kilograms (kg) divided by the square of height in meters (m). BMI-standard deviation score (SDS) (also known as BMI z-score), which was adjusted for child’s age and sex based on local growth chart, was used to interpret physical development and growth in children and adolescents [29]. Body fat percentage was assessed by bioelectrical impedance analysis (BIA) using Tanita body composition analyzer (Model BC-418). Blood pressure was measured using Carescape V100 Dinamap. Fasting blood samples were obtained and assayed for fasting glucose, fasting insulin and lipids. Blood glucose was also measured at 2 h after the subjects underwent an oral glucose tolerance test (OGTT) by consuming a drink consisting of 75 g glucose. Homeostatic model assessment for insulin resistance (HOMA-IR) was calculated as previously described [30].

### 2.3. Assessment of Dietary Intake

Dietary intake was assessed using a 3-day food diary, which is completed by each participant or their caregiver if the child is below the age of 12 years old. The participant or caregiver of the participant was asked to record the participant’s intake of meals, snacks, beverages and supplements for a period of three days. The food diary required participants to include the name of the food or beverage consumed, ingredients used in the meal, amount consumed (portion sizes), preparation methods (type of oil, cooking method) and brand name (if available). Pictures illustrating the portion size or amount of food or beverage were included in the food diary to guide the participants in estimating the amount of food or beverage consumed. A study team member would explain to the participant or their caregiver on how to fill up the food diary, which also contained instructions and examples of food records. The participants were asked to send pictures of the food that they consumed to the study team. A well-trained nutritionist reviewed and verified the written 3-day food diaries with the food pictures sent. The intake of energy, macronutrients, micronutrients and food under the different food groups (in terms of quantity/amount) were analyzed using a nutritional analysis program: Dietplan, Forestfield Software, UK (Version 7.00.62), which consists of a local database of energy and nutrient composition of food, as well as food label information of food products obtained from local stores [31]. Macronutrients were energy adjusted using the nutrient density method [32] and expressed as percentage (%) of total energy (kcal). Micronutrients were expressed as per 1000 kcal of energy: (amount of micronutrient/total energy (kcal))*1000 kcal.

The acceptable macronutrient distribution range (AMDR) for carbohydrates (%kcal), total fat (%kcal), saturated fat (%kcal), protein (%kcal) and recommended dietary allowance (RDA) for dietary fiber (g/1000 kcal) were obtained from Dietary Guidelines for Americans 2020–2025 [33]. The RDA for calcium (mcg), iron (mcg) and vitamin A (mcg) were obtained from Health Promotion Board Singapore dietary guidelines [34]. Appendix A showed the AMDR for macronutrients and RDA for micronutrients for the various age groups.

The reported foods consumed by the participants were categorized into nine main food groups: deep fried food, fast food and processed convenience food, fish, fruits, savory snacks, sugar-sweetened beverage (SSB), sweet snacks, vegetables and whole grains. Appendix A lists the food items under each of the food groups.

### 2.4. Evaluation of Eating Behavior

The Three-Factor Eating Questionnaire—Revised 18-item version (TFEQ-R18) is a validated questionnaire comprised of three subscales: cognitive dietary restraint, emotional eating and uncontrolled eating [35,36]. The 18 items are on a 4-point response scale, e.g., definitely true/mostly true/mostly false/definitely false, and a score between 1 and 4 is assigned to each response. Item scores are summated into the three subscales and higher scores in the respective subscales are indicative of greater cognitive dietary restraint, emotional or uncontrolled eating.

### 2.5. Classification of MHO and MUO Children/Adolescents

There is currently a lack of consensus on the definition of MHO [37]. However, most studies have defined MHO by either absence of metabolic syndrome (having ≤2 criteria) or total absence of metabolic abnormalities [37,38]. Children/adolescents with obesity were classified as having MHO or MUO according to two different definitions, metabolic syndrome (MS) and metabolic health (MH), in accordance with a standard protocol [39]. The criteria for both MS and MH definitions are adapted and modified from the International Diabetes Federation (IDF) consensus definition of metabolic syndrome in children and adolescents [40]: (1) hypertriglyceridemia: fasting triglycerides ≥ 1.7 mmol/L or on hyperlipidemia medication, (2) dyslipidemia: high-density lipoprotein (HDL) cholesterol < 1.03 mmol/L for children under age of 16, HDL < 1.03mmol/L for male ≥ 16 years old and HDL < 1.29 mmol/L for female ≥ 16 years old, (3) abnormal glucose tolerance: fasting glucose ≥ 5.6 mmol/L or glucose at 2 h OGTT ≥ 7.8 mmol/L or on diabetic medication, (4) elevated blood pressure: blood pressure ≥ 90th percentile based on age, sex and height or on hypertensive medication [41]. In this study, for the MS definition, MHO was considered as being obese with fewer than two of the criteria. For the MH definition, MHO was considered as being obese without any of the criteria.

### 2.6. Statistical Analysis

All analyses were performed using SPSS 27.0 and STATA 17.0 with level of significance set at 2-sided *p* < 0.05. Our data did not follow a normal distribution so non-parametric statistical methods were used for data analyses. Descriptive statistics for numerical and categorical variables were presented as median (interquartile range: 25th percentile–75th percentile) and proportion (%), respectively. Differences in clinical characteristics between children/adolescents with MHO and children/adolescents with MUO were analyzed by Mann–Whitney U test for continuous parameters and Chi-square for categorical parameters. Kruskal–Wallis H test was used to analyze the difference in food groups, nutrient intakes and eating behavior across the 3 ethnic groups. Quantile regression was performed to model median differences in food groups, nutrient intakes and eating behavior between children/adolescents with MHO and children/adolescents with MUO, and to analyze the association between food groups/nutrient intake/eating behavior and continuous metabolic parameters with adjustment for age, sex, race and BMI-SDS. Logistic regression was performed to identify the factors (food groups/nutrients/eating behavior) associated with various metabolic conditions and MUO phenotype, with adjustment for age, sex, race and BMI-SDS.

## 3. Results

### 3.1. Clinical Characteristics of Participants

There were no significant differences in demographics data such as age, sex, race and monthly household income, and adiposity outcomes between children/adolescents with MHO and children/adolescents with MUO for both the MS and MH definitions (Table 1).

With regards to the MS definition, children/adolescents with MUO had significantly higher systolic blood pressure (median: 130 mmHg (interquartile range: 124–134 mmHg) vs. 118 mmHg (111–126 mmHg), *p* = 0.003), triglycerides levels (1.55 mmol/L (1.10–2.17 mmHg) vs. 1.01 mmol/L (0.89–1.25 mmHg), *p* = 0.014) and lower HDL cholesterol (1.01 mmol/L (0.89–1.17 mmol/L) vs. 1.17 mmol/L (0.99–1.25 mmol/L), *p* = 0.048) compared to children/adolescents with MHO.

With regards to the MH definition, children/adolescents with MUO had higher systolic blood pressure (125 mmHg (115–132 mmHg) vs. 114 mmHg (108–118 mmHg), *p* = 0.004), fasting glucose (4.90 mmol/L (4.60–5.20 mmHg) vs. 4.70 mmHg (4.50–4.80 mmHg), *p* = 0.027), fasting insulin (24.7 mU/L (16.9–33.0 mU/L) vs. 19.7 mU/L (13.0–26.3 mU/L), *p* = 0.039), HOMA-IR (5.15 (3.30–7.73) vs. 4.26 (2.72–5.49), *p* = 0.019) and lower HDL cholesterol (1.06 mmol/L (0.95–1.19 mmol/L) vs. 1.24 mmol/L (1.19–1.29 mmol/L), *p* = 0.002) levels compared to children/adolescents with MHO.

### 3.2. Food Groups, Nutrient Intakes and Eating Behavior between Children/adolescents with MHO and Children/Adolescents with MUO

With regards to the MH definition, children/adolescents with MUO were found to consume a significantly lower amount of whole grains (0.00 (0.00–0.00 g) vs. 18.5 g (0.00–69.8 g), *p* = 0.027) and polyunsaturated fat (6.26% kcal (5.17–7.45% kcal) vs. 6.92% kcal (5.85–9.02% kcal), *p* = 0.027), and displayed lower cognitive restraint in eating (15.0 (13.0–17.0) vs. 16.0 (14.0–19.0), *p* = 0.009) compared to children/adolescents with MHO (Table 2).

With regards to the MS definition, there were no significant differences in food groups, nutrients intake, eating behavior and percentage of participants meeting the AMDR and RDA of nutrients between children/adolescents with MHO and children/adolescents with MUO (Appendix A).

### 3.3. Food Groups, Nutrient Intakes and Eating Behavior between Children/Adolescents with MHO and Children/Adolescents with MUO Stratified by Sex or Race

Gender differences [42,43] and ethnicity [44,45] have been reported to influence dietary intakes. Hence, we further stratified the children/adolescents with obesity by sex or race, and examined the nutrient intakes between children/adolescents with MHO and children/adolescents with MUO.

With regards to the MH definition, male children/adolescents with MUO consumed a significantly lower amount of fruits (0.00 g (0.00–17.9 g) vs. 23.3 g (0.00–48.9 g), *p* = 0.010) and reported lower cognitive dietary restraint (14.0 (13.0–16.0) vs. 15.5 (13.8–19.0), *p* = 0.031) compared to male children/adolescents with MHO. Female children/adolescents with MUO were found to consume a significantly lower amount of whole grains (0.00 g (0.00–0.00 g) vs. 133 g, *p* < 0.001), and exhibited higher emotional (7.00 (5.00–8.00) vs. 3.00, *p* = 0.043) and uncontrolled eating (21.0 (18.0–23.0) vs. 17.0, *p* = 0.027) compared to female children/adolescents with MHO (Table 3).

With regards to the MS definition, there were no significant differences in food groups, nutrients intake and eating behavior between children/adolescents with MHO and children/adolescents with MUO stratified by sex (Appendix A).

There were significant differences in vegetables, carbohydrates and protein between the three ethnic groups. However, due to the small sample size of Indian children/adolescents (*n* = 3) within the cohort, comparisons were made between Chinese and Malay children/adolescents. There were significant differences in vegetable, carbohydrate, protein, total fat and saturated fat intakes between Chinese and Malay children/adolescents (Appendix A).

With regards to the MH definition, among the Chinese participants, children/adolescents with MUO had a significantly lower SSB intake (237 mL (108–336 mL) vs. 333 mL, *p* = 0.020) and higher monounsaturated fat intake (14.1% kcal (13.2–15.1% kcal) vs. 9.88% kcal, *p* = 0.002) than children/adolescents with MHO. Among the Malay participants, children/adolescents with MUO had lower polyunsaturated fat intake (5.84% kcal (5.17–6.92% kcal) vs. 8.38% kcal (6.03–9.35% kcal), *p* = 0.039), and reported higher cognitive dietary restraint (16.0 (13.0–18.0) vs. 15.5 (14.0–19.0), *p* = 0.016) compared to children/adolescents with MHO (Table 4).

With regards to the MS definition, there were no significant differences in food groups, nutrients intake and eating behavior between children/adolescents with MHO and children/adolescents with MUO stratified by race (Appendix A).

### 3.4. Association between Food Groups/Nutrients/Eating Behavior and Risk Factors of Metabolic Syndrome

Iron intake was found to be negatively associated with HDL cholesterol (β:−2.34, 95%CI: −4.65- −0.04) and glucose level at 2 h OGTT (β:−0.37, 95%CI: −0.67–−0.07), while deep fried food, processed food and convenience food intakes were positively associated with both systolic (β: 2.84, 95%CI: 0.95–6.62) and diastolic (β: 4.83, 95%CI: 0.61–9.04) blood pressure (Table 5).

We also examined the association between food groups/nutrients/eating behavior and metabolic conditions. Protein (OR= 0.791, 95%CI: 0.642–0.974), calcium (OR= 0.991, 95%CI: 0.982–1.000) and iron (OR= 0.527, 95%CI: 0.309–0.899) intakes, and cognitive dietary restraint (OR: 0.711, 95%CI: 0.523–0.966) were associated with a lower risk of elevated blood pressure. Iron intake was associated with a higher risk of dyslipidemia in HDL cholesterol (OR= 2.363, 95%CI: 1.258–4.437) but a lower risk of abnormal glucose tolerance (OR= 0.349, 95%CI: 0.134–0.908). Polyunsaturated fat intake (OR= 0.529, 95%CI: 0.284–0.986) and cognitive dietary restraint (OR: 0.681, 95%CI: 0.472–0.984) were associated with a lower risk of MUO by the MH definition (Table 6).

## 4. Discussion

Our findings demonstrated variations in dietary intakes between children/adolescents with MHO and children/adolescents with MUO for different MHO definitions. Significant differences in dietary factors between children/adolescents with MHO and children/adolescents with MUO were reported with the more stringent MH definition [39]. The lack of significant differences for the MS definition may be due to the heterogeneity of the definition as some children/adolescents with MHO under the MS definition would have one metabolic abnormality that may overall be similar to the MUO phenotype [37]. Hence, our results highlighted the importance of establishing a consensus definition for MHO [37,38,46]. Children/adolescents with MHO were found to consume more whole grains and polyunsaturated fats in their diet, and had higher cognitive dietary restraint than children/adolescents with MUO. Whole grains are rich in dietary fiber, vitamins, minerals and beneficial phytochemicals from plants [47], and they are recommended in dietary guidelines for healthy eating [48,49]. An increased consumption of whole grains was associated with a lower risk of metabolic syndrome [50], and subjects with MHO were found to have a higher intake of whole grains compared to subjects with MUO [20]. Polyunsaturated fats are considered the good fats, which help to reduce bad cholesterol (LDL) in the blood and lower cardiovascular risk [51], and they were shown to ameliorate obesity and obesity-induced metabolic syndrome [52]. Telle-Hansen et al. also reported lower levels of total polyunsaturated fatty acid in subjects with MUO compared to subjects with MHO [53]. There were no significant differences in the proportion of participants meeting the recommended intake of macronutrients and micronutrients between children/adolescents with MHO and children/adolescents with MUO. Our cohort of children/adolescents with obesity had a higher intake of saturated fat and lower intake of micronutrients including calcium, dietary fiber, iron and vitamin A than recommended amounts [33]. Cognitive dietary restraint is the perceived effort to limit dietary intake [54] and higher cognitive dietary restraint is associated with a reduction in adiposity outcomes [55]. Our observation of higher cognitive dietary restraint among children/adolescents with MHO may indicate an intentional dietary restriction to improve metabolic health outcomes. We did not find significant differences in energy intake and other macro- and micronutrient intakes between children/adolescents with MHO and children/adolescents with MUO. Although this lack of significant differences was also observed in other studies [56,57], our small sample size of children/adolescents with MHO and MUO phenotypes may have contributed to the lack of statistical significant difference in the dietary intakes.

We found that dietary intakes of fruits and whole grains, and eating behavior were significantly different between children/adolescents with MHO and children/adolescents with MUO stratified by sex, and our findings are consistent with other studies that reported gender difference in dietary intakes and eating behavior [58,59]. There were also significant differences in SSB, monounsaturated fat and polyunsaturated fat intakes as well as cognitive dietary restraint between children/adolescents with MHO and children/adolescents with MUO stratified by race. This supports our aim to investigate dietary intakes and eating behavior between MHO and MUO phenotypes in our local cohort of children/adolescents due to the differing diets between different ethnic groups [44,45]. However, the stratification analyses by sex and race were based on a small sample size of children/adolescents with MHO and MUO phenotypes.

Similar to previous reports [60,61,62,63,64], we found that deep fried food, fast food and processed convenience food intakes were positively associated with blood pressure [60,61,62], while the intake of micronutrients such as calcium and iron was negatively associated with metabolic outcomes [63,64]. Protein intake was also found to be associated with a lower risk of hypertension [65]. However, only polyunsaturated fat intake and cognitive dietary constraint were shown to be associated with metabolic health in our cohort of children/adolescents with obesity.

The susceptibility to obesity-related comorbidities among individuals with obesity who are exposed to the same obesogenic environment (i.e., our cohort of children/adolescents with MHO and children/adolescents with MUO reported no significant differences in most food groups and nutrient intakes except whole grains and polyunsaturated fat intake) might be attributed to the interaction between genetic variants and diet (also known as nutrigenetics), which affects the body’s response to specific nutrients [66]. Significant interactions between genetic variants of lipid metabolism genes and dietary intakes of fat were found to be associated with blood lipid profiles in adults with overweight and obesity [67]. The greater risk of developing metabolic syndrome conferred by the GG genotype of the leptin receptor genetic variant (rs3790433) was shown to be abrogated among individuals with a high intake of polyunsaturated fatty acids [68]. In addition, there are other aspects of diet and nutrition that have not been investigated in our study. The eating habits such as tendency to snack, eating in the absence of hunger, and the number, duration and regularity of meals have been found to be associated with metabolic health in individuals with obesity [27]. Accumulating evidence has indicated a role of diet in regulating the gut microbiome, which has been proposed as an underlying mechanism in obesity and metabolic diseases [69]. A Western-style diet that is high in fat and refined carbohydrates may promote pro-inflammatory intestinal bacteria that are linked to obesity and metabolic diseases [70]. Hence, to better elucidate the impact of diet on obesity and its associated metabolic health, it is imperative to examine the other factors that interact with dietary intake and response.

There are severable limitations in this study. Firstly, our sample size is small and the study may be underpowered to establish significant differences between children/adolescents with MHO and children/adolescents with MUO. Secondly, our dataset may not have included all the important micronutrients, e.g., polyphenols, which may influence the metabolic phenotype of individuals with obesity [71]. Thirdly, a longitudinal study is required to establish the effect of polyunsaturated fat intake and cognitive dietary restraint on the metabolic health of individuals with obesity.

Despite the caveats, our study had several strengths. We had a well-phenotyped cohort of children/adolescents with obesity and this allowed us to clearly categorize the cohort into the MHO and MUO phenotypes. The dietary intakes of the participants were collected in the form of a comprehensive 3-day food diary, which was reported to have better agreement with observed food intakes [72]. Children/adolescents with MHO were fairly well matched with children/adolescents with MUO in terms of demographics such as age, sex, race, household income and adiposity measures such as BMI, BMI-SDS, waist to hip ratio and body fat percentage, which were confounding variables of metabolic health in obesity. Hence, this allowed a fair comparison of dietary intakes between children/adolescents with MHO and children/adolescents with MUO.

## 5. Conclusions

In conclusion, our study demonstrated that a healthier diet composition and positive eating behavior may contribute to favorable metabolic outcomes in children/adolescents with obesity. Interventions targeting the dietary intake of polyunsaturated fats and eating behavior such as cognitive dietary restraint may improve metabolic health in children/adolescents with obesity.

## Figures and Tables

**Table 1 nutrients-14-04796-t001:** Clinical characteristics between children/adolescents with MHO and children/adolescents with MUO classified by MS and MH definitions.

Parameter	All (*n* = 52)	MHO (*n* = 42)	MUO (*n* = 10)	*p*	MHO (*n* = 12)	MUO (*n* = 40)	*p*
Age (years)	14.1 (12.3–16.1)	14.1 (11.8–16.1)	14.1 (13.8–16.3)	0.531	12.6 (8.59–17.0)	14.6 (13.1–16.1)	0.182
Sex (%Male/%Female)	59.6/40.4	60/40	60/40	1.000	83/17	52.5/47.5	0.093
Race (%Chinese/%Malay/%Indian)	40.4/53.8/5.8	38/55/7	50/50/0	0.597	25/67/8	45/50/5	0.457
Monthly household income < SGD 2000 (%)	15.6	15	17	1.000	10	18	1.000
BMI (kg/m^2^)	35.9 (32.1–41.3)	35.9 (31.8–40.9)	36.4 (32.3–44.0)	0.763	33.6 (30.1–39.4)	36.5 (32.4–42.2)	0.152
BMI-SDS	2.43 (2.16–2.64)	2.44 (2.17–2.63)	2.28 (2.05–2.79)	0.781	2.36 (1.99–2.55)	2.44 (2.19–2.68)	0.422
Waist to hip ratio	0.98 (0.93–1.02)	0.98 (0.95–1.02)	0.95 (0.91–1.00)	0.189	1.01 (0.94–1.03)	0.98 (0.93–1.00)	0.142
Body fat percentage (%)	48.2 (39.4–55.4)	47.9 (39.8–55.7)	50.3 (35.5–54.8)	0.952	51.5 (37.1–62.6)	48.0 (40.1–54.3)	0.558
Systolic blood pressure (mmHg)	120 (111–130)	118 (110–126)	130 (124–134)	0.003 *	113 (108–118)	124 (115–132)	0.004 *
Diastolic blood pressure (mmHg)	66 (59–72)	64 (57–72)	70 (62–73)	0.197	58 (55–72)	67 (60–73)	0.080
Total cholesterol (mmol/L)	4.49 (3.92–5.04)	4.52 (4.03–5.03)	4.18 (3.54–5.17)	0.493	4.61 (3.77–5.05)	4.37 (3.98–5.04)	0.991
Triglycerides (mmol/L)	1.12 (0.94–1.35)	1.01 (0.86–1.24)	1.55 (1.10–2.17)	0.014 *	0.88 (0.69–1.24)	1.13 (0.95–1.42)	0.059
HDL cholesterol (mmol/L)	1.11 (0.98–1.24)	1.17 (0.99–1.25)	1.01 (0.89–1.17)	0.048 *	1.24 (1.19–1.29)	1.06 (0.95–1.19)	0.002 *
LDL cholesterol (mmol/L)	2.84 (2.33–3.17)	2.95 (2.40–3.38)	2.52 (2.06–2.95)	0.099	2.96 (2.32–3.17)	2.80 (2.33–3.32)	0.871
Fasting glucose (mmol/L)	4.75 (4.60–5.10)	4.70 (4.60–5.00)	5.00 (4.58–5.20)	0.434	4.65 (4.50–4.78)	4.85 (4.60–5.20)	0.027 *
Glucose at 2 h of OGTT (mmol/L)	5.50 (5.00–6.20)	5.30 (4.88–6.10)	5.85 (5.23–8.55)	0.140	5.30 (5.10–5.68)	5.55 (4.85–6.38)	0.535
Fasting insulin (mU/L)	22.6 (14.3–30.7)	21.4 (14.1–30.4)	25.0 (19.1–38.5)	0.403	17.4 (11.4–25.1)	25.0 (17.2–32.5)	0.039 *
HOMA-IR	4.85 (3.14–6.43)	4.48 (3.09–6.29)	5.41 (3.76–8.84)	0.410	3.67 (2.30–5.22)	5.15 (3.35–7.70)	0.019 *

Data were presented as median (interquartile range: 25th–75th percentile) and percentage (%) for continuous and categorical variables, respectively. Differences in continuous variables between groups were analyzed using Mann–Whitney U test, while differences in categorical variables between groups were analyzed using Chi-square test. Asterisk * denotes significance of *p* < 0.05.

**Table 2 nutrients-14-04796-t002:** Food groups, nutrients intakes and eating behavior between children/adolescents with MHO and children/adolescents with MUO by MH definition.

	MH Definition		
	MHO (*n* = 12)	MUO (*n* = 40)	*p*
**Food groups (continuous variables)**			
Deep fried food (g)	76.6 (19.0–136)	55.1 (39.3–129)	0.558
Fast food and processed convenience food (g)	121 (16.1–152)	54.1 (0.00–135)	0.502
Fish (g)	0.00 (0.00–0.00)	0.00 (0.00–66.7)	0.788
Fruits (g)	17.7 (0.00–44.4)	0.00 (0.00–39.8)	0.721
Savory snacks (g)	1.22 (0.00–21.3)	5.00 (0.00–34.5)	0.965
Sugar-sweetened beverage, SSB (ml)	342 (163–421)	278 (161–464)	0.417
Sweet snacks (g)	52.0 (6.25–118)	23.5 (0.00–58.2)	0.171
Vegetables (g)	73.5 (43.7–113)	82.3 (37.5–140)	0.430
Whole grains (g)	18.5 (0.00–69.8)	0.00 (0.00–0.00)	0.027 *
**Nutrients (continuous variables)**			
Total energy (kcal)	1856 (1670–2470)	1855 (1730–2260)	0.239
Carbohydrates (%kcal)	49.1 (46.8–52.1)	45.3 (40.2–51.9)	0.539
Protein (%kcal)	16.4 (14.7–17.6)	17.7 (15.4–21.4)	0.536
Total fat (%kcal)	34.4 (32.8–37.2)	36.0 (31.8–39.5)	0.918
Saturated fat (%kcal)	12.6 (10.1–14.2)	12.3 (11.1–14.3)	0.319
Monounsaturated fat (%kcal)	11.7 (10.4–12.5)	13.5 (11.4–14.9)	0.851
Polyunsaturated fat (%kcal)	6.92 (5.85–9.02)	6.26 (5.17–7.45)	0.027 *
Beta-carotene (mcg per 1000 kcal)	5.16 (0.00–50.1)	0.19 (0.00–3.83)	0.655
Calcium (mg per 1000 kcal)	304 (183–368)	252 (209–298)	0.166
Cholesterol (mg per 1000 kcal)	172 (101–229)	198 (149–228)	0.160
Dietary fiber (g per 1000 kcal)	6.73 (5.85–7.59)	6.60 (5.98–7.82)	0.955
Iron (mg per 1000 kcal)	6.54 (5.02–7.21)	6.03 (5.24–7.09)	0.719
Sodium (mg per 1000 kcal)	1550 (1410–1840)	1780 (1360–2050)	0.839
Vitamin A (mcg per 1000 kcal)	248 (114–338)	258 (187–363)	0.797
**% of participants meeting AMDR/RDA of nutrients**			
Carbohydrates † (AMDR)	91.7	50	0.186
Total fat † (AMDR)	58.3	42.5	0.924
Saturated fat † (AMDR)	16.7	17.5	0.689
Protein † (AMDR)	100	100	1.000
Calcium ‡ (RDA)	8.3	5	0.498
Dietary fiber † (RDA)	8.3	5	0.891
Iron ‡ (RDA)	83.3	50	0.290
Vitamin A ‡ (RDA)	50	20	0.072
**Eating behavior (continuous variables)**			
Cognitive dietary restraint	16.0 (14.0–19.0)	15.0 (13.0–17.0)	0.009 *
Emotional eating	6.00 (3.25–6.00)	6.00 (4.00–8.00)	1.000
Uncontrolled eating	22.0 (18.0–24.5)	21.0 (19.0–24.0)	0.766

Data were presented as median (interquartile range: 25th–75th percentile) and percentage (%) for continuous and categorical variables, respectively. Differences in continuous variables between groups were analyzed using quantile regression with adjustment for age, sex, race and BMI-SDS, while differences in categorical variables between groups were analyzed using logistic regression with adjustment for age, sex, race and BMI-SDS. Asterisk * denotes significance of *p* < 0.05. † AMDR and RDA of nutrients were according to Dietary Guidelines for Americans 2020–2025, ‡ RDA of nutrients were according to dietary guidelines by Health Promotion Board, Singapore.

**Table 3 nutrients-14-04796-t003:** Food groups, nutrient intakes and eating behavior between children/adolescents with MHO and children/adolescents with MUO (MH definition) stratified by sex.

	MH Definition					
	Male			Female		
	MHO (*n* = 10)	MUO (*n* = 21)	*p*	MHO (*n* = 2)	MUO (n = 19)	*p*
**Food groups**						
Deep fried food (g)	53.4 (7.18–110)	80.9 (43.3–138)	0.997	204	43.8 (23.3–106)	0.401
Fast food and processed convenience food (g)	121 (19.5–150)	75.0 (0.00–151)	0.855	107	50.0 (0.00–133)	0.182
Fish (g)	0.00 (0.00–0.00)	14.0 (0.00–95.5)	0.577	0.00	0.00 (0.00–47.7)	0.774
Fruits (g)	23.3 (0.00–48.9)	0.00 (0.00–17.9)	0.010 *	13.1	4.00 (0.00–60.0)	0.970
Savory snacks (g)	1.22 (0.00–19.8)	0.00 (0.00–33.1)	0.940	16.7	12.8 (0.00–44.5)	0.742
Sugar-sweetened beverage, SSB (ml)	355 (209–472)	257 (129–472)	0.163	182	313 (207–444)	0.212
Sweet snacks (g)	52.0 (18.8–121)	16.7 (0.00–50.8)	0.097	61.8	26.7 (7.67–66.7)	0.202
Vegetables (g)	66.1 (39.8–118)	71.3 (35.9–112)	0.725	86.3	85.3 (36.7–159)	0.555
Whole grains (g)	17.2 (0.00–38.6)	0.00 (0.00–42.3)	0.933	133	0.00 (0.00–0.00)	<0.001 *
**Nutrients**						
Total energy (kcal)	1860 (1670–2460)	2110 (1790–2420)	0.253	2080	1780 (1680–1900)	0.432
Carbohydrates (% kcal)	49.1 (47.1–52.7)	49.0 (38.6–52.8)	0.900	43.1	44.0 (41.1–50.6)	0.052
Protein (% kcal)	16.0 (14.5–17.3)	17.6 (15.9–21.5)	0.987	19.4	19.7 (14.0–21.4)	0.396
Total fat (% kcal)	34.4 (32.9–36.4)	34.7 (30.8–39.7)	0.409	37.5	36.2 (32.7–39.6)	0.971
Saturated fat (% kcal)	12.3 (10.0–13.7)	11.6 (9.93–13.8)	0.371	14.3	12.7 (12.1–14.9)	0.386
Monounsaturated fat (% kcal)	11.7 (10.0–12.2)	13.5 (11.4–15.1)	0.786	13.5	13.4 (9.77–14.5)	0.244
Polyunsaturated fat (% kcal)	7.47 (5.46–9.25)	6.09 (5.11–7.80)	0.268	6.82	6.56 (5.67–7.24)	0.746
Beta-carotene (mcg per 1000 kcal)	3.39 (0.00–59.3)	0.00 (0.00–1.32)	0.548	15.4	0.81 (0.00–27.6)	0.801
Calcium (mg per 1000 kcal)	298 (158–318)	225 (195–284)	0.367	383	277 (221–359)	0.282
Cholesterol (mg per 1000 kcal)	161 (83.7–185)	208 (160–254)	0.097	305	190 (136–219)	0.096
Dietary fiber (g per 1000 kcal)	6.73 (5.72–7.80)	6.67 (5.85–8.14)	0.397	6.93	6.44 (5.97–7.53)	0.982
Iron (mg per 1000 kcal)	6.54 (4.94–7.14)	6.01 (5.24–7.29)	0.367	7.12	6.55 (5.23–6.92)	0.540
Sodium (mg per 1000 kcal)	1520 (1290–1840)	1760 (1480–2050)	0.795	1680	1890 (1340–2060)	0.957
Vitamin A (mcg per 1000 kcal)	188 (108–300)	219 (123–353)	0.275	395	314 (227–402)	0.348
**Eating behavior**						
Cognitive dietary restraint	15.5 (13.8–19.0)	14.0 (13.0–16.0)	0.031 *	17.5	17.0 (14.0–18.0)	0.999
Emotional eating	6.00 (5.50–6.50)	5.00 (3.50–6.00)	0.490	3.00	7.00 (5.00–8.00)	0.043 *
Uncontrolled eating	23.0 (18.0–25.0)	21.0 (19.0–24.0)	0.772	17.0	21.0 (18.0–23.0)	0.027 *

Data were presented as median (interquartile range: 25th–75th percentile). Due to small sample size for female MHO children/adolescents (*n* = 2), no interquartile range is available. Differences in continuous variables between groups were analyzed using quantile regression with adjustment for age, race and BMI-SDS. Asterisk * denotes significance of *p* < 0.05.

**Table 4 nutrients-14-04796-t004:** Food groups, nutrient intakes and eating behavior between children/adolescents with MHO and children/adolescents with MUO (MH definition) stratified by race.

	MH Definition					
	Chinese			Malay		
	MHO (*n* = 3)	MUO (*n* = 18)	*p*	MHO (*n* = 8)	MUO (*n* = 20)	*p*
**Food groups**						
Deep fried food (g)	47.3	78.8 (42.3–161)	0.972	78.3 (19.2–151)	48.8 (23.5–119)	0.274
Fast food and processed convenience food (g)	0.00	54.1 (0.00–123)	0.389	129 (35.7–198)	75.3 (0.00–170)	0.296
Fish (g)	0.00	0.00 (0.00–105)	1.000	0.00 (0.00–0.00)	5.00 (0.00–66.7)	0.771
Fruits (g)	26.1	7.07 (0.00–62.1)	0.963	4.58 (0.00–52.2)	0.00 (0.00–29.5)	1.000
Savory snacks (g)	0.00	6.42 (0.00–30.2)	0.551	15.8 (0.61–30.5)	0.00 (0.00–44.9)	0.683
Sugar-sweetened beverage, SSB (ml)	333	237 (108–336)	0.020 *	291 (57.5–378)	370 (219–588)	0.194
Sweet snacks (g)	124	23.5 (5.75–63.7)	0.750	34.3 (6.25–90.7)	22.5 (0.00–48.9)	0.422
Vegetables (g)	70.5	125 (53.7–169)	0.387	66.1 (43.7–101)	50.1 (29.9–92.0)	0.466
Whole grains (g)	19.0	0.00 (0.00–0.00)	0.680	17.2 (0.00–69.8)	0.00 (0.00–15.9)	0.209
**Nutrients**						
Total energy (kcal)	1670	1860 (1770–2300)	0.830	1860 (1680–2400)	1820 (1680–2220)	0.358
Carbohydrates (% kcal)	52.6	40.5 (36.3–43.9)	0.051	47.6 (45.7–49.5)	50.6 (46.0–53.2)	0.076
Protein (% kcal)	14.6	21.3 (17.1–23.2)	0.224	16.9 (15.8–18.7)	15.8 (14.0–19.6)	0.211
Total fat (% kcal)	32.8	39.1 (35.7–40.7)	0.058	35.7 (34.3–37.9)	32.6 (30.6–36.2)	0.076
Saturated fat (% kcal)	13.0	12.4 (11.9–14.9)	0.922	10.9 (9.84–14.9)	11.8 (10.1–13.2)	0.226
Monounsaturated fat (% kcal)	9.88	14.1 (13.2–15.1)	0.002 *	12.0 (11.6–14.3)	12.1 (9.90–14.2)	0.991
Polyunsaturated fat (% kcal)	6.46	6.83 (6.00–7.96)	0.215	8.38 (6.03–9.35)	5.84 (5.17–6.92)	0.039 *
Beta-carotene (mcg per 1000 kcal)	24.9	0.09 (0.00–1.50)	0.066	3.39 (0.00–45.3)	0.27 (0.00–4.86)	0.872
Calcium (mg per 1000 kcal)	316	233 (199–288)	0.585	298 (151–368)	259 (205–299)	0.332
Cholesterol (mg per 1000 kcal)	85.7	213 (179–248)	0.236	176 (158–229)	198 (116–222)	0.484
Dietary fiber (g per 1000 kcal)	7.39	6.22 (5.83–8.10)	0.712	6.67 (5.56–7.49)	7.08 (6.03–7.71)	0.148
Iron (mg per 1000 kcal)	6.50	5.83 (5.24–6.70)	0.735	6.84 (5.26–7.21)	6.72 (5.06–7.34)	0.365
Sodium (mg per 1000 kcal)	964	1860 (1370–2560)	0.352	1630 (1440–1840)	1700 (1390–2050)	0.834
Vitamin A (mcg per 1000 kcal)	286	288 (195–403)	0.997	248 (121–338)	250 (155–325)	0.468
**Eating behavior**						
Cognitive dietary restraint	15.0	14.0 (13.0–17.0)	0.612	15.5 (14.0–19.0)	16.0 (13.0–18.0)	0.016 *
Emotional eating	3.00	5.50 (3.75–8.00)	0.085	6.00 (6.00–7.50)	6.00 (4.00–7.75)	0.398
Uncontrolled eating	19.0	21.0 (18.0–25.0)	0.110	23.0 (18.8–25.0)	21.0 (19.0–23.0)	0.549

Data were presented as median (interquartile range: 25th–75th percentile). Due to small sample size for Chinese MHO children/adolescents (*n* = 3), no interquartile range is available. Differences in continuous variables between groups were analyzed using quantile regression with adjustment for age, sex and BMI-SDS. Asterisk * denotes significance of *p* < 0.05.

**Table 5 nutrients-14-04796-t005:** Association between food groups/nutrients/eating behavior and continuous metabolic parameters.

	BMI-SDS		Triglycerides		HDL Cholesterol		Fasting Glucose		Glucose at 2 h OGTT		Systolic Blood Pressure		Diastolic Blood Pressure	
	β	95% CI	β	95% CI	β	95% CI	β	95% CI	β	95% CI	β	95% CI	β	95% CI
**Food groups**														
Deep fried food (g)	6.01	−83.5–95.5	−0.85	−47.1–45.4	82.0	−55.6–220	−7.16	−60.6–46.3	−0.16	−17.0–16.7	2.84 *	0.95–6.62 *	0.84	−2.93–4.61
Fast food and processed convenience food (g)	−17.0	−140–106	−31.8	−92.9–29.3	−35.5	−248–177	7.25	−67.8–82.3	1.86	−21.2–25.0	1.02	−2.89–4.92	4.83*	0.61–9.04 *
Fish (g)	0.00	−52.3–52.3	0.00	−28.9–28.9	−23.5	−114–67.1	0.00	−32.1–32.1	7.39	−2.32–17.1	0.00	−1.78–1.78	0.00	−2.05–2.05
Fruits (g)	7.72	−24.1–39.6	0.22	−15.9–16.3	6.00	−47.2–59.2	−6.83	−24.7–11.0	−0.80	−6.97–5.37	−0.05	−1.13–1.02	0.19	−1.05–1.42
Savory snacks (g)	16.6	−12.8–46.0	−0.50	−16.4–15.4	−14.7	−69.0–39.6	−9.14	−28.0–9.72	−1.81	−7.61–3.99	−0.04	−1.05–0.97	0.95	−0.25–2.15
Sugar-sweetened beverage, SSB (ml)	−79.8	−322–163	−28.5	−164–107	133	−370–635	31.5	−112–175	−25.9	−72.8–21.1	−1.33	−9.85–7.19	−3.73	−12.9–5.44
Sweet snacks (g)	7.37	−43.7–58.3	0.27	−26.6–27.1	24.9	−59.9–110	0.35	−30.9–31.6	2.02	−7.74–11.8	−0.02	−1.61–1.57	−0.02	−1.86–1.82
Vegetables (g)	38.3	−38.4–115	−9.54	−48.7–29.6	−89.9	−212–32.0	5.74	−40.1–51.6	15.6	−2.98–28.1	−1.31	−3.65–1.03	−1.02	−4.00–1.95
Whole grains (g)	0.00	−14.4–14.4	0.00	−7.64–7.64	0.00	−45.8–45.8	0.00	−6.94–6.94	0.00	−2.64–2.64	0.00	−0.83–0.83	0.00	−1.02–1.02
**Macronutrients**														
Carbohydrates (% kcal)	−2.09	−9.80–5.63	1.58	−2.22–5.39	9.57	−3.09–16.06	2.42	−2.18–7.01	0.91	−0.42–2.23	0.15	−0.12–0.42	0.09	−0.19–0.37
Protein (% kcal)	2.19	−2.44–6.82	−0.73	−3.08–1.62	−7.25	−14.8–0.31	−1.31	−4.12–1.51	0.38	−0.50–1.26	−0.08	−0.23–0.07	−0.03	−0.20–0.15
Total fat (% kcal)	2.48	−3.65–8.62	−1.39	−4.67–1.88	−1.49	−11.6–8.64	0.06	−3.60–3.72	−0.41	−1.52–0.69	0.01	−0.17–0.20	0.01	−0.21–0.23
Saturated fat (% kcal)	0.39	−2.66–3.44	−0.48	−2.17–1.22	−1.51	−6.10–3.07	0.15	−1.76–2.06	−0.20	−0.73–0.33	−0.01	−0.11–0.08	−0.02	−0.13–0.09
Monounsaturated fat (% kcal)	1.29	−1.77–4.35	1.26	−0.35–2.86	−0.90	−6.01–4.22	1.11	−0.61–2.83	−0.28	−0.91–0.36	−0.05	−0.16–0.06	−0.02	−0.14–0.10
Polyunsaturated fat (% kcal)	−0.078	−1.89–1.73	−0.57	−1.53–0.39	2.81	−0.74–6.35	−0.19	−1.29–0.91	−0.05	−0.44–0.33	−0.00	−0.06–0.06	0.05	−0.02–0.12
**Micronutrients**														
Beta-carotene (mcg per 1000 kcal)	−0.00	−16.2–16.2	−0.41	−8.86–8.03	0.00	−27.7–27.7	−1.72	−11.4–7.99	−0.32	−3.50–2.86	−0.01	−0.65–0.64	0.00	−0.83–0.83
Calcium (mg per 1000 kcal)	−10.6	−116–94.7	−9.17	−64.0–45.6	127	−36.1–290	−6.82	−71.2–57.5	−10.7	−31.1–9.69	−2.40	−5.98–1.18	−1.08	−5.03–2.87
Cholesterol (mg per 1000 kcal)	−10.0	−91.2–71.1	−10.4	−52.9–32.2	−62.7	−213–87.2	−17.4	−68.1–33.3	−6.91	−23.1–9.32	0.94	−2.07–3.95	−0.30	−3.37–2.78
Dietary fiber (g per 1000 kcal)	−0.33	−2.36–1.69	0.06	−0.87–0.98	−0.87	−4.32–2.59	−0.12	−1.30–1.07	−0.09	−0.45–0.26	−0.01	−0.08–0.06	0.02	−0.07–0.10
Iron (mg per 1000 kcal)	−0.89	−2.09–0.91	0.36	−0.37–1.09	−2.34 *	−4.65–−0.04 *	−0.17	−1.09–0.76	−0.37 *	−0.67–−0.07 *	−0.04	−0.08–0.01	−0.02	−0.07–0.04
Sodium (mg per 1000 kcal)	207	−507–921	−34.7	−372–303	576	−519–1672	165	−272–601	−59.7	−196–77	−7.49	−31.4–16.4	−2.69	−27.5–22.1
Vitamin A (mcg per 1000 kcal)	−60.1	−210–90.3	−43.8	−120–32.7	−62.8	−328–202	−58.1	−142–25.7	−13.2	−40.0–13.6	−0.80	−6.10–4.51	−1.48	−7.40–4.44
**Eating behavior**														
Cognitive dietary restraint	−0.63	−3.87–2.61	0.15	−1.65–1.95	1.29	−3.91–6.49	−0.06	−2.06–1.95	0.09	−0.53–0.71	−0.06	−0.17–0.04	0.02	−0.11–0.15
Emotional eating	−0.00	−2.54–2.54	0.00	−1.43–1.43	−0.00	−4.44–4.44	0.00	−1.51–1.51	0.00	−0.48–0.48	−0.00	−0.09–0.09	0.05	−0.05–0.15
Uncontrolled eating	1.62	−2.55–5.78	−0.98	−3.11–1.15	4.15	−3.09–11.4	−0.74	−3.20–1.71	−0.32	−1.07–0.44	−0.07	−0.19–0.06	0.05	−0.11–0.21

Data for the total sample size (*n* = 52) were presented as β, 95% confidence interval (CI). Association between food groups/nutrients/eating behavior and metabolic parameters (except BMI-SDS) was analyzed using quantile regression with adjustment for age, sex, race and BMI-SDS (for BMI-SDS, analysis was adjusted for age, sex and race only). Asterisk * denotes significance of *p* < 0.05.

**Table 6 nutrients-14-04796-t006:** Nutritional factors predictive of metabolic conditions.

	Elevated Blood Pressure		Hypertriglyceridemia		Dyslipidemia (HDL)		Abnormal Glucose Tolerance		MUO (MS Definition)		MUO (MH Definition)	
	OR	95% CI	OR	95% CI	OR	95% CI	OR	95% CI	OR	95% CI	OR	95% CI
**Food groups**												
Deep fried food (g)	1.005	0.997–1.013	1.003	0.993–1.014	0.997	0.989–1.005	0.994	0.980–1.007	1.000	0.991–1.009	0.996	0.986–1.006
Fast food and processed convenience food (g)	1.002	0.995–1.008	0.992	0.980–1.005	0.999	0.993–1.006	0.994	0.981–1.007	0.998	0.989–1.006	0.997	0.989–1.005
Fish (g)	1.007	0.994–1.020	1.010	0.992–1.029	0.998	0.984–1.013	1.013	0.996–1.031	1.009	0.994–1.023	1.028	0.999–1.059
Fruits (g)	1.007	0.992–1.021	1.001	0.978–1.024	0.992	0.977–1.007	0.993	0.973–1.013	1.001	0.985–1.018	1.004	0.982–1.026
Savory snacks (g)	1.010	0.992–1.028	0.991	0.959–1.024	1.003	0.984–1.022	1.002	0.976–1.029	1.014	0.993–1.034	1.019	0.980–1.059
Sugar-sweetened beverage, SSB (ml)	1.002	1.000–1.005	1.001	0.998–1.003	1.000	0.997–1.002	0.994	0.987–1.001	1.002	0.999–1.004	1.001	0.998–1.004
Sweet snacks (g)	0.997	0.986–1.009	0.994	0.975–1.014	0.996	0.983–1.009	0.986	0.960–1.014	0.995	0.978–1.011	0.988	0.974–1.002
Vegetables (g)	0.997	0.990–1.004	0.988	0.970–1.007	1.001	0.994–1.008	1.004	0.997–1.012	1.001	0.993–1.008	0.998	0.990–1.007
Whole grains (g)	0.991	0.975–1.007	1.006	0.990–1.023	1.004	0.990–1.018	0.996	0.976–1.015	1.006	0.992–1.021	0.993	0.978–1.008
**Macronutrients**												
Carbohydrates (% kcal)	1.100	0.982–1.232	1.022	0.880–1.187	0.913	0.811–1.028	0.994	(0.864–1.142)	1.020	0.905–1.150	0.988	0.861–1.134
Protein (% kcal)	0.791 *	0.642–0.974 *	0.980	0.741–1.297	1.115	0.929–1.340	1.213	(0.955–1.541)	0.942	0.764–1.160	1.074	0.830–1.391
Total fat (% kcal)	0.976	0.848–1.122	0.972	0.799–1.182	1.083	0.935–1.253	0.871	(0.706–1.074)	1.002	0.852–1.179	0.983	0.814–1.188
Saturated fat (% kcal)	1.057	0.795–1.405	0.918	0.618–1.365	0.986	0.725–1.341	0.903	(0.581–1.403)	1.125	0.807–1.570	0.799	0.555–1.153
Monounsaturated fat (% kcal)	0.981	0.772–1.247	1.353	0.909–2.012	1.004	0.780–1.291	1.009	(0.700–1.455)	1.146	0.843–1.558	1.133	0.841–1.526
Polyunsaturated fat (% kcal)	0.870	0.598–1.265	0.957	0.584–1.568	0.914	0.613–1.363	0.623	(0.318–1.222)	0.803	0.502–1.284	0.529 *	0.284–0.986 *
**Micronutrients**												
Beta-carotene (mcg per 1000 kcal)	0.977	0.939–1.017	0.360	0.070–1.859	1.000	0.995–1.005	0.984	(0.924–1.048)	0.982	0.925–1.042	0.995	0.989–1.001
Calcium (mg per 1000 kcal)	0.991 *	0.982–1.000 *	0.997	0.987–1.007	1.002	0.995–1.009	0.999	(0.989–1.009)	0.992	0.981–1.003	0.997	0.989–1.005
Cholesterol (mg per 1000 kcal)	1.001	0.992–1.010	1.009	0.995–1.022	1.007	0.997–1.017	1.006	(0.994–1.018)	1.005	0.994–1.015	1.003	0.992–1.014
Dietary fiber (g per 1000 kcal)	0.714	0.473–1.079	0.967	0.584–1.601	1.188	0.863–1.635	0.747	(0.416–1.340)	0.655	0.365–1.175	1.015	0.702–1.468
Iron (mg per 1000 kcal)	0.527 *	0.309–0.899 *	1.033	0.583–1.832	2.363 *	1.258–4.437 *	0.349 *	(0.134–0.908) *	0.716	0.416–1.231	1.154	0.643–2.071
Sodium (mg per 1000 kcal)	1.000	0.999–1.001	1.000	0.999–1.002	0.999	0.998–1.000	1.000	(0.999–1.002)	0.999	0.998–1.001	1.001	0.999–1.003
Vitamin A (mcg per 1000 kcal)	0.998	0.993–1.002	0.996	0.989–1.003	1.001	0.997–1.005	1.002	(0.996–1.008)	0.998	0.992–1.003	1.000	0.995–1.004
**Eating behavior**												
Cognitive dietary restraint	0.711 *	0.523–0.966 *	0.987	0.690–1.413	0.747	0.527–1.059	1.259	0.853–1.859	0.797	0.557–1.139	0.681 *	0.472–0.984 *
Emotional eating	0.866	0.641–1.172	1.017	0.695–1.488	0.958	0.698–1.314	0.968	0.650–1.442	0.877	0.617–1.246	1.084	0.756–1.553
Uncontrolled eating	0.972	0.805–1.174	0.987	0.763–1.275	1.027	0.840–1.256	0.934	0.720–1.213	0.925	0.745–1.148	1.109	0.873–1.410

Data for the total sample size (*n* = 52) were presented odds ratio (OR), 95%CI. Association between food groups/nutrients/eating behavior and metabolic abnormalities was analyzed using logistic regression with adjustment for age, sex, race and BMI-SDS. Asterisk * denotes significance of *p* < 0.05.

## Data Availability

The data presented in this study are available within the manuscript and Appendix A.

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
