# Peer review of "Dietary Intakes and Eating Behavior between Metabolically Healthy and Unhealthy Obesity Phenotypes in Asian Children and Adolescents"

_nutrients, 2022, doi:10.3390/nu14224796_

Round 1
Reviewer 1 Report
Dear authors, thank you for submitting the manuscript entitled “Dietary intakes and eating behavior between metabolically 2 healthy and unhealthy obesity phenotypes in Asian children” to Nutrients. The research question is important in the field of obesity research in childhood and adolescents and here interesting and relevant data are presented, although there are some issues that needs to be addressed.
Abstract – add total number of participating children, and a blank or never a blank should be used between number and unit – the same style should be used
Line 99: ethical issue: it is written that consent was obtained from all study participants – was there also written consent from parents? Add the information to the manuscript
Line 102 for children`s BMI the use of appropriate BMI-percentiles for age (or Z-Scores) should be used for calculations and interpretation in regard to physical development– this should be added
Line 119 how was the amount of intakes estimated? Were there instructions/pictures e.g. for the children themselves or their caregivers? More information should be included within the text
Line 141 sweet crunchy cereals or muesli/ sweet spreads are important foods that are known to be heavily consumed by children and are not mentioned in S2 – the same is for alternative cereal products e.g. amaranth, millet… are these foods of minor importance in this group in Asia – please explain, how were the portions calculated?
Line 143 who filled out these questionnaires – it is important to distinct between the younger children with less cognitive abilities compared to the older ones. This issue should be addressed in the method section
Line 186 p-values should be added within the comparison
Overall the number of children in the MHO group (by MH definition) is very (too) low for interpretation and further distribution in groups should not be performed - alternative calculation possibilities e.g. regression analysis are recommended.
The difference between MS and MH definition should be discussed in more detail already in the methods but importantly further in the discussion section. Add evidence from the scientific literature
Table 1 add a column on total sample size e.g. all participating children/adolescents with obesity
Tab 3 for further stratification of according to sex the number of children is too low – find other way to explore gender influence e.g. correlation and/or regressionanalysis,
Tab 5/6 –information on participant numbers should be added– maybe better to use linear regression for both or explain the use of the actual statistical methods clearly e.g. quantile regression– and further explain within the discussion the protein intake (not only the cognitive dietary restraint). Furthermore, it is unclear why different statistical approaches are used for more or less the same titles for both tables (ending in associations) appropriate information should be added to the titles. And information on these approaches should be worked in more appropriate in statistics section.
Line 304 information on actual intakes vs. recommended should be given and it should be discussed if the group with higher consumption is reaching recommendations for the discussed nutrients
Citations should be double checked – doi numbers are sometimes missing, sometimes a link and sometimes text only
Overall, it is an interesting work on childhood obesity, which is a very important field to tackle obesity and it is difficult to reach sufficient numbers for statistical analysis. Although a careful handling of the data is important to not overinterpret the results. It is suggested to stay with the possible and meaningful analysis and put the group/categorial variables e.g. gender, culture as covariate or confounder but omit group comparisons with such low numbers. Furthermore, it is suggested to include a table on comparing the actual intake with the recommendations for the children/adolescents. Per definition 19year old are not children anymore, especially the term adolescents should be included – this should be adopted throughout the manuscript.
Discussion should be more carefully written with regards to the low numbers and mentioned more often, not only within the limitation section.
Author Response
Author's reply to review report (Reviewer report 1)
We sincerely thank the reviewers for their constructive comments and suggestions to improve the manuscript. We have revised the manuscript according to the reviewers’ comments and suggestions, and the revisions are highlighted in yellow.
Comments and Suggestions for Authors
Dear authors, thank you for submitting the manuscript entitled “Dietary intakes and eating behavior between metabolically 2 healthy and unhealthy obesity phenotypes in Asian children” to Nutrients. The research question is important in the field of obesity research in childhood and adolescents and here interesting and relevant data are presented, although there are some issues that needs to be addressed.
Abstract – add total number of participating children, and a blank or never a blank should be used between number and unit – the same style should be used
Response: We have now indicated the total number of participants (n=52) and revised the style between number and unit as suggested by the reviewer (line 26).
Line 99: ethical issue: it is written that consent was obtained from all study participants – was there also written consent from parents? Add the information to the manuscript
Response: We thank the reviewer for the suggestion. All study participants (children with obesity) were below the age of 21 years when they were recruited. Hence, written consent was obtained from the children and their parents or legal guardian. We have now indicated that written informed consent was obtained from all study participants and their parents or legal guardian (lines 108-109).
Line 102 for children`s BMI the use of appropriate BMI-percentiles for age (or Z-Scores) should be used for calculations and interpretation in regard to physical development– this should be added
Response: We agree with the reviewer that appropriate BMI-percentiles for age (or Z-scores) should be used for calculations and interpretation in regards to physical development in children and adolescents. Hence, we have previously indicated that BMI and BMI-SDS were calculated. We have now indicated that BMI-SDS (also known as BMI z-score) which was adjusted for child’s age and sex based on local growth chart was used to interpret physical development and growth in children and adolescents (lines 114-116).
Line 119 how was the amount of intakes estimated? Were there instructions/pictures e.g. for the children themselves or their caregivers? More information should be included within the text
Response: We thank the reviewer for the comment. The amount of intakes were estimated based on pictures of portion size, instructions and examples provided in the food diary. We have now included these information within the text (lines 132-136).
Line 141 sweet crunchy cereals or muesli/ sweet spreads are important foods that are known to be heavily consumed by children and are not mentioned in S2 – the same is for alternative cereal products e.g. amaranth, millet… are these foods of minor importance in this group in Asia – please explain, how were the portions calculated?
Response: We thank the reviewer for the comment. For sweet crunchy cereals or sweet spreads, they would be classified under the “Sweet snacks” food group. For muesli or alternative cereal e.g. millet, they would be classified under the “Whole gains” food group. However, based on the food diary records, we did not observe any of the participants consuming these products. We have now included that the quantity of food intake under the different food groups (in terms of quantity/amount) were analyzed using a nutritional analysis program: Dietplan, Forestfield Software, UK (Version 7.00.62), which consists of a local database of energy and nutrient composition of food, as well as food label information of food products obtained from local stores (lines 138-141).
Line 143 who filled out these questionnaires – it is important to distinct between the younger children with less cognitive abilities compared to the older ones. This issue should be addressed in the method section
Response: We thank the reviewer for the comment and we agree with the reviewer that it is important to distinct between the younger and older children who are filling up the questionnaire. We have stated that the 3-day food diary is completed by each participant or their caregiver if the child is below the age of 12 years old (lines 126-127).
Line 186 p-values should be added within the comparison
Response: We have now added the p-values within the comparisons (lines 210-222, lines 230-234, lines 251-258, lines 267-273).
Overall the number of children in the MHO group (by MH definition) is very (too) low for interpretation and further distribution in groups should not be performed - alternative calculation possibilities e.g. regression analysis are recommended.
Response: We agree with the reviewer that the sample size is small, and we have stated the small sample size as a limitation of this study (lines 365-366, line 395). However, due to gender and ethnic group differences in food/nutrition intake, we have performed subgroup analyses between the gender groups, and between the ethnic groups. We have analyzed the data using quantile regression analysis.
The difference between MS and MH definition should be discussed in more detail already in the methods but importantly further in the discussion section. Add evidence from the scientific literature
Response: We thank the reviewer for the comment. We have now included more discussion on the MS and MH definitions in the methods (lines 169-171) and discussion sections (lines 327-329).
Table 1 add a column on total sample size e.g. all participating children/adolescents with obesity
Response: We thank the reviewer for the comment. We have now included a column to indicate the median values of the metabolic parameters in the total sample size of 52 children/adolescents with obesity (Table 1).
Tab 3 for further stratification of according to sex the number of children is too low – find other way to explore gender influence e.g. correlation and/or regression analysis,
Response: We thank the reviewer for the comment. We acknowledged that the sample size is too small for stratification analysis by sex and race. However, as sex and race are reported to influence dietary intakes, we performed regression analysis to explore gender and race influence. The regression analysis used in our study is quantile analysis (instead of linear regression) due to our data not being normally distributed (lines 188-189).
Tab 5/6 –information on participant numbers should be added– maybe better to use linear regression for both or explain the use of the actual statistical methods clearly e.g. quantile regression– and further explain within the discussion the protein intake (not only the cognitive dietary restraint). Furthermore, it is unclear why different statistical approaches are used for more or less the same titles for both tables (ending in associations) appropriate information should be added to the titles. And information on these approaches should be worked in more appropriate in statistics section.
Response: We thank the reviewer for the comment. We have now added the information on participant number (n=52) and explained the use of non-parametric statistical method e.g. Mann-Whitney U test, Kruskal-Wallis H test, quantile regression for data analyses in this study. In Table 5, we used quantile regression to examine the associations between dietary intakes and continuous metabolic parameters. In Table 6, we used logistic regression to examine dietary actors predictive of the binary outcome of the metabolic disease. We have now revised the titles for Tables 5 and 6. We have provided discussion on cognitive dietary restraint because the aim of our study was to examine factors influencing MHO/MUO phenotype in our children/adolescents with obesity and cognitive dietary restraint (not protein intake) was found to be associated with MUO phenotype (Table 6). However, we have now indicated in the discussion that protein intake was associated with lower risk of hypertension (lines 370-371)
Line 304 information on actual intakes vs. recommended should be given and it should be discussed if the group with higher consumption is reaching recommendations for the discussed nutrients
Response: We thank the reviewer for the comment. We have now included the actual intake vs. recommended amounts in the discussion section (lines 341-346).
Citations should be double checked – doi numbers are sometimes missing, sometimes a link and sometimes text only
Response: We apologize for the formatting error, and we have now corrected the formatting for the citations.
Overall, it is an interesting work on childhood obesity, which is a very important field to tackle obesity and it is difficult to reach sufficient numbers for statistical analysis. Although a careful handling of the data is important to not overinterpret the results. It is suggested to stay with the possible and meaningful analysis and put the group/categorial variables e.g. gender, culture as covariate or confounder but omit group comparisons with such low numbers. Furthermore, it is suggested to include a table on comparing the actual intake with the recommendations for the children/adolescents. Per definition 19year old are not children anymore, especially the term adolescents should be included – this should be adopted throughout the manuscript.
Response: We thank the reviewer for the positive and constructive comments. We agree with the reviewer that one of the caveats for our study is the small sample size (n=52). All statistical analysis performed for this manuscript was adjusted for age, sex, ethnicity and BMI-SDS which are considered as confounding variables for metabolic health in obesity. Despite our small sample size, we have included the subgroup analysis/comparisons for sex and race which are important factors influencing dietary intake. Table 2 showed the comparisons between actual intake and recommendations of macro- and micro-nutrients, and Supplementary Table 1 showed the recommended intake for the dietary macro- and micro-nutrients. We have now adopted the term adolescents throughout the manuscript.
Discussion should be more carefully written with regards to the low numbers and mentioned more often, not only within the limitation section.
Response: We agree with the reviewer and we have now mentioned the small sample size throughout the discussion section of the manuscript.

Reviewer 2 Report
Please address the following concerns:
1. Abrupt transitions in the introduction section after reference 15 and 24.
2. Please check the spacing between > and numbers throughout the document and be consistent.
3. On line 112, please say 'in accordance with a standard protocol' instead of previously described.
4. On line 116, please use past tense as this is not an ongoing study.
5. On line 153, abrupt introduction of MS and MH in the paper and not sure if those qualifiers are adding much to the study.
6. Please be consistent in the use of double parenthesis for lines 187-195.
7. On line 220, please say 'children with MUO'.
8. What is the advantage of categorizing using MS & MH definitions in addition to the use of MHO/MUO categorization; makes the paper confusing to read and results hard to follow.
9. On lines 235-240, results appear to be in an opposite direction to what was expected.
10. On line 257, please omit 'and' before processed food.
11. Page numbering is messed up after page 9.
12. Lines 297- 302, very confusing and difficult to follow.
13. On line 298, do you mean MH definition instead of MHO definition?
14. There is no mention of strengths of the study in the discussion section.
15. Very brief conclusion and no discussion of the implications of this study's findings.
16. The study authors would have reached the same conclusion without adding MS & MH categories in the study design.
Author Response
Author’s reply to the review report (Reviewer report 2)
We sincerely thank the reviewers for their constructive comments and suggestions to improve the manuscript. We have revised the manuscript according to the reviewers’ comments and suggestions, and the revisions ae highlighted in yellow.
Comments and Suggestions for Authors
Please address the following concerns:
- Abrupt transitions in the introduction section after reference 15 and 24.
Response: We thank the reviewer for the comment. For reference 15, we introduced the importance of diet in contributing to obesity and obesity-related metabolic morbidities. Thus, after reference 15, we cited published studies (provide examples) that showed the different dietary nutrients/components that were associated with obesity and metabolic health in obesity. For reference 24, we introduced that apart from diet, eating behaviour is also associated with obesity. Thus, after reference 24, we cited published studies (provided examples) that showed the association between eating behaviour and obesity/obesity-related metabolic morbidities.
- Please check the spacing between > and numbers throughout the document and be consistent.
Response: We have now standardized the spacing between the symbol (<, >, ≤ or ≥) and number throughout the manuscript.
- On line 112, please say 'in accordance with a standard protocol' instead of previously described.
Response: We have now revised the sentence as suggested by the reviewer (line 173).
- On line 116, please use past tense as this is not an ongoing study.
Response: We have now revised the sentence to past tense form as suggested by the reviewer (lines 182-184).
- On line 153, abrupt introduction of MS and MH in the paper and not sure if those qualifiers are adding much to the study.
Response: We thank the reviewer for the comment. We have now revised the sentences to introduce the MS and MH definitions in the methods section (lines 169-171).
- Please be consistent in the use of double parenthesis for lines 187-195.
Response: We have now standardized the use of parenthesis for lines and throughout the manuscript.
- On line 220, please say 'children with MUO'.
Response: We apologize for the error and we have now revised to “children with MUO” (line 250).
- What is the advantage of categorizing using MS & MH definitions in addition to the use of MHO/MUO categorization; makes the paper confusing to read and results hard to follow.
Response: We thank the reviewer for the comment. There is currently a lack of consensus in the definition of MHO so we have decided to compare MHO and MUO using both MS and MH definitions which are the 2 commonly used MHO definitions in published data (lines 327-329).
- On lines 235-240, results appear to be in an opposite direction to what was expected.
Response: We agree with the reviewer that the results appeared to be in an opposite direction to what was expected. While we were not able to explain this unexpected trend, we have highlighted in the discussion section that, genetics and gut microbiome may influence the nutrient metabolism which in turn affect the metabolic phenotype in obesity (lines 374-394) i.e. children/adolescents with MHO may have favourable genetic factors and gut microbiome that allowed efficient glucose metabolism, resulting in maintenance of insulin sensitivity.
- On line 257, please omit 'and' before processed food.
Response: We have now omitted “and” before processed food (line 294).
- Page numbering is messed up after page 9.
Response: We apologize for the error in page numbering. We have now corrected the page numbering of the manuscript.
- Lines 297- 302, very confusing and difficult to follow.
Response: We apologize for the confusion and we have now revised the sentences (lines 323-331).
- On line 298, do you mean MH definition instead of MHO definition?
Response: We initially wanted to convey that MH definition is the more stringent definition for MHO in this study but we agree with the reviewer that our point was confusing and unclear. We have now revised to “MH definition” instead of “MHO definition (MH definition)” (line 326).
- There is no mention of strengths of the study in the discussion section.
Response: We have now mentioned the strengths of the study in the discussion section (lines 402-411).
- Very brief conclusion and no discussion of the implications of this study's findings.
Response: We thank the reviewer for the comment. We have now discussed the implications of this study’s findings (lines 415-417).
- The study authors would have reached the same conclusion without adding MS & MH categories in the study design.
Response: We agree with the reviewer that we would have reached the same conclusion by using the MH definition alone. However, there is currently a lack of consensus in the definition of MHO so we have decided to compare MHO and MUO using both MS and MH definitions which are the 2 commonly used MHO definitions in published data (lines 169-171, lines 327-329).
